DANNET: deep attention neural network for efficient ear identification in biometrics

http://orcid.org/0000-0002-6927-3168 Alex Deepthy Mary 1 deepthy.alex@mangalam.in
M. Kalpana Chowdary 2
Mengash Hanan Abdullah 3
M. Venkata Dasu 4
http://orcid.org/0000-0003-3678-9229 Kryvinska Natalia 5
J. Chinna Babu 4 jchinnababu@gmail.com
Kiran Ajmeera 2
1 Department of Electronics and Communication Engineering, Mangalam College of Engineering , Ettumanoor, Kerala , India
2 Department of Computer Science and Engineering, MLR Institute of Technology , Hyderabad, Telangana , India
3 Department of Information Systems, College of Computer and Information Sciences, Princess Nourah bint Abdulrahman University , Riyadh , Saudi Arabia
4 Department of Electronics and Communication Engineering, Annamacharya University , Rajampet, Andhra Pradesh , India
5 Department of Information Management and Business Systems, Faculty of Management, Comenius University Bratislava , Bratislava , Slovakia
Raza Khalid
Electronic publication date: 2024 Dec 18
Publication date: 2024
Volume: 10
Electronic Location ID: e2603
Received 2024 May 7; Accepted 2024 Nov 21
Copyright: © 2024 Alex et al.
Copyright year: 2024
Copyright holder: Alex et al.
License: This is an open access article distributed under the terms of the Creative Commons Attribution License, which permits unrestricted use, distribution, reproduction and adaptation in any medium and for any purpose provided that it is properly attributed. For attribution, the original author(s), title, publication source (PeerJ Computer Science) and either DOI or URL of the article must be cited.
License URL: https://creativecommons.org/licenses/by/4.0/

Keywords: Ear biometrics, Deep learning, Segmentation, YSegNet, Ensemble, UNet

Funding: Princess Nourah bint Abdulrahman University, Riyadh, Saudi Arabia PNURSP2024R114 Department of Information Management and Business Systems, Faculty of Management, Comenius University Bratislava, Slovakia This work was supported by the Princess Nourah bint Abdulrahman University Researchers Supporting Project number (PNURSP2024R114), Princess Nourah bint Abdulrahman University, Riyadh, Saudi Arabia and the Department of Information Management and Business Systems, Faculty of Management, Comenius University Bratislava, Slovakia. The funders had no role in study design, data collection and analysis, decision to publish, or preparation of the manuscript.

==============================
Biometric identification, particularly ear biometrics, has gained prominence amidst the global prevalence of mask-wearing, exacerbated by the COVID-19 outbreak. This shift has highlighted the need for reliable biometric systems that can function effectively even when facial features are partially obscured. Despite numerous proposed convolutional neural network (CNN) based deep learning techniques for ear detection, achieving the expected efficiency and accuracy remains a challenge. In this manuscript, we propose a sophisticated method for ear biometric identification, named the encoder-decoder deep learning ensemble technique incorporating attention blocks. This innovative approach leverages the strengths of encoder-decoder architectures and attention mechanisms to enhance the precision and reliability of ear detection and segmentation. Specifically, our method employs an ensemble of two YSegNets, which significantly improves the performance over a single YSegNet. The use of an ensemble method is crucial in ear biometrics due to the variability and complexity of ear shapes and the potential for partial occlusions. By combining the outputs of two YSegNets, our approach can capture a wider range of features and reduce the risk of false positives and negatives, leading to more robust and accurate segmentation results. Experimental validation of the proposed method was conducted using a combination of data from the EarVN1.0, AMI, and Human Face datasets. The results demonstrate the effectiveness of our approach, achieving a segmentation framework accuracy of 98.93%. This high level of accuracy underscores the potential of our method for practical applications in biometric identification. The proposed innovative method demonstrates significant potential for individual recognition, particularly in scenarios involving large gatherings. When complemented by an effective surveillance system, our method can contribute to improved security and identification processes in public spaces. This research not only advances the field of ear biometrics but also provides a viable solution for biometric identification in the context of mask-wearing and other facial obstructions.

Introduction

The surge in interest insecure automated identity systems has led to a growth in research within several domains such as intelligent systems and computer vision. Due to their consistency across time, ease of acquisition, and individuality, biometrics are used by the majority of human identification systems. The most widely used biometrics for human identification include facial features, iris scans, fingerprints, palmprints, hand geometry, voice, and signatures. While the voice of an individual is categorized as a blend of physiological and biometric traits, researchers have developed numerous technologies to differentiate diverse biometric traits. These applications span a range of purposes, including forensic investigations and security measures. During and after the COVID-19 pandemic, biometrics such as facial recognition and fingerprints were restricted to some extent due to mask-wearing and concerns about disease transmission. However, in such situations ear biometrics proves efficient and promising. The ear biometric is difficult to forge and the ear shape is also highly unique to each individual so it is more preferable than other ways.

In a fetus, the auditory development begins around 20 weeks of gestation (Graven & Browne, 2008). Between 4 months and 8 years of age the outer structure of the ear develops (Hall, 2000). The structure of the human outer ear is provided by Benzaoui et al. (2023) for reference. The structure of each individual’s ears is unique. Thus, ear biometric identification techniques are more advantageous for automated recognition as they do not require user cooperation, are contactless, and are non-intrusive. In applications such as surveillance, where facial recognition faces challenges like occlusion or low-resolution images, integrating ear identification can serve as an additional source of information for more robust recognition (Pan et al., 2008; Kisku et al., 2009). Additionally, ear biometrics holds promise in various commercial applications in today’s world. However, to leverage ear biometrics effectively, the initial step involves accurate detection or identification of the ear. Ongoing research in this field continues to explore novel techniques and architectures for enhancing the accuracy and efficiency of ear detection and recognition systems.

Over time, numerous researchers have developed diverse techniques for the detection and segmentation of ears, as evidenced by the mentioned works (Wahab, Hemayed & Fayek, 2012; Srivastava, Agrawal & Bansal, 2020; Prakash & Gupta, 2012a). A few researchers have experimented the scope of artificial intelligence in the area of ear detection, includes both machine learning and deep learning. In the survey by Pflug & Busch (2012) most of the existing ear detecting techniques and its challenges are explained. In Burge & Burger (2000), detection of ears is done using deformable contours. The technique uses canny detector to identify edges followed by edge relaxation algorithm. Due to initial user intervention for contour initialization the technique is semi-automated. The authors have used ear shape as a factor to detect ears based on morphological operations and fourier descriptors in Kumar & Wu (2012). The technique is automated, but the efficiency of the technique was not adequate. The authors of Prakash & Gupta (2012b) used skin color and graph matching for localizing of the ears but the technique is invariant to variations, rotation and shape. Researchers have explained various proposed techniques in Abaza et al. (2013) for detecting ears. Particle swarm optimization (PSO) and sequential search algorithms were also experimented by the authors in Chandy et al. (2020) for improvisation of ear localization but efficiency were not met as expected. Speeded up robust features (SURF) features were used for ear identification using K-nearest neighbor (KNN) and convolutional neural network (CNN) in Nanaware et al. (2019). Another approach presented by Eyiokur, Yaman & Ekenel (2018) involves the use of two CNNs combined to identify front and side views of the ears, achieving an accuracy of 84%. Further, an ensemble of three CNNs are proposed to detect ears where the output is the weighted average output from each CNN (Ganapathi et al., 2020). In Cintas et al. (2017), a proposed deep learning model utilizes CNN with geometric morphometrics. A multiple scale faster R-CNN (region-based CNN) for detecting ears in an uncontrolled environment is proposed in Zhang & Mu (2017). An encoder-decoder framework is also proposed in (Emeršič et al., 2017a) for classifying ear and non-ear. DenseNet models were also experimented and explained in Zhang et al. (2018). Ears recognition is carried out using residual neural network (ResNet) in Emeršič et al. (2017b), the vanishing gradient problem is addressed in this model. Few researchers have also presented detailed reviews on ear biometrics in Kamboj, Rani & Nigam (2022), Wang, Yang & Zhu (2021). Further, in Oyebiyi et al. (2023) a comprehensive review of various algorithms and trends in ear biometrics are mentioned but lacks an in-depth analysis of the practical challenges associated with deploying ear biometric systems in dynamic environments, such as varying head angles or occlusions. Mehta & Singh (2022) propose the use of ensemble learning techniques to improve the accuracy of ear biometric systems while (Mahajan & Singla, 2024) a hybrid of CNN and a self-attention mechanism with a gated recurrent unit (GRU). Both techniques significantly raise the complexity and computational demands, which can be a drawback for real-time applications.

All the fore mentioned literatures have demonstrated using any one dataset or any particular scenario that makes their model efficient to a specific task. In this work the proposed work is trained and tested using a dataset that combines three datasets to make the model perform well. Open-source databases make it easier for researchers to experiment their ideas and also validate and assess their proposed works with the existing ones. Over the years several ear datasets have been available to researchers with easy access. Some of the relevant ear datasets which are mentioned in Benzaoui et al. (2023) is categorised as constrained and unconstrained databases based on the complexity. In order for the model to work efficiently in all scenarios, a dataset with both constrained and unconstrained images are considered. The dataset employed here is developed by incorporating imagery from the following databases: i) AMI Ear Database: The Images that are using for Mathematical Analysis (AMI) Ear Database is a freely accessible collection comprising 700 ear images captured in various poses from 100 subjects. Each subject contributes one left ear image and six right ear images to the database. These AMI images are acquired in an closed indoor atmosphere from the department of computer science of ULPGC, Las Palmas, Spain (Gonzalez, Alvarez & Mazorra CTIM, 2012). The ages of the subjects ranged from 19–65 years.

ii) EarVN1.0. Dataset: The dataset sums to 28,412 images consisting of the right and left ears of 164 Asian population collected in 2018 (Hoang, 2019). These images were collected from unconstrained conditions that includes lighting and camera system conditions. The ear images in various pose, illumination and scale were cropped from the original facial image to create the dataset.

iii) Human Face (Ear Detection with Annotation) Dataset: This collection comprises pictures of human faces with marked ear regions. It encompasses 440 images, each potentially featuring one or more human faces. This dataset is invaluable for training computer vision models focused on ear detection in human faces. The images were sourced from various origins, leading to differing resolutions and lighting conditions. Additionally, the dataset is accompanied by a CSV file containing annotations for the ear regions within each image.

The proposed approach employs an encoder-decoder framework, achieving success due to its simple yet robust architecture, effective training with limited datasets, multi-scale feature extraction and fusion, and its ability to generate high-quality segmentation maps. The primary contribution of this article is the introduction of an advanced deep ensemble network, which combines two YSegNet models to accurately detect ears for a variety of applications. The rest of the article is organized as follows: “Methodology” describes the methodology of the proposed ensemble, “Results and Discussions” outlines the results and discussion that includes performance metrics used and evaluation and “Conclusion” concludes the article.

Methodology

This section outlines the methodology used to develop the proposed ensemble model for segmentation. The model integrates two YSegNets, (Alex et al., 2022) each built upon the UNet (Ronneberger, Fischer & Brox, 2015) architecture, to enhance segmentation accuracy and robustness. By leveraging the strengths of both YSegNets and incorporating advanced techniques like attention-gated skip connections and dense blocks, the model achieves superior performance. The following subsections describe the ensemble model architecture, the rationale behind its design, and the integration of the two YSegNet models.

Ensemble model overview

The proposed ensemble model integrates two YSegNets, each with a unique configuration to enhance segmentation accuracy. The first YSegNet utilizes a pretrained VGG-16 as the encoder and incorporates dense blocks to improve feature extraction and reuse throughout the network. Dense blocks allow for better gradient flow and prevent vanishing gradients, which is crucial for learning deep features effectively.

The second YSegNet deviates from the traditional structure by taking input directly and combining with output of the first YSegNet, instead of using a separate pretrained VGG-16 encoder. This setup enables the second YSegNet to process enhanced feature representations from the first YSegNet. Additionally, the second YSegNet employs attention blocks in the skip connections, allowing it to focus on the most salient features while filtering out irrelevant information. This attention mechanism enhances the feature transfer between the encoder and decoder, further refining the segmentation accuracy.

The rationale for this ensemble design is to capitalize on the complementary strengths of the two YSegNets. The first YSegNet, with its dense blocks, is highly effective at capturing detailed hierarchical features, while the second YSegNet, utilizing attention blocks, fine-tunes these features by concentrating on critical regions. By combining the outputs of both networks, the model delivers more accurate and robust segmentation results. The integration of these two architectures creates a synergy that boosts performance in handling complex segmentation tasks. The flow diagram and the schematic representation of the proposed model is given in Figs. 1 and 2, respectively.

Figure 1 Illustration of proposed ensemble for ear segmentation.

Proposed Ensemble used to represent ear segmentation.

Figure 2 Architecture of the proposed ensemble model.

Base model

YSegNet 1: The YSegNet model integrates VGG-16 as its encoder and employs dense blocks as skip connections to enhance its segmentation capabilities. By leveraging the VGG-16 network, YSegNet benefits from a robust and deep feature extraction backbone. VGG-16, pre-trained on extensive datasets, provides a strong foundation for capturing hierarchical and detailed features from input images. Its series of convolutional and pooling layers progressively downsample the input, effectively extracting rich, multi-scale features necessary for accurate segmentation.

In addition to the VGG-16 encoder, YSegNet incorporates dense blocks specifically as skip connections between the encoder and decoder. These dense blocks serve to facilitate the fusion of features from various layers of the network. Unlike traditional architectures where dense blocks are part of the encoder, their role here is to enhance the combination of high-resolution details with the features processed by VGG-16. Dense blocks concatenate outputs from multiple preceding layers, improving the integration and utilization of spatial information. This approach enhances the flow of information and ensures that detailed features are effectively preserved and utilized during the segmentation process.

The decoder in YSegNet utilizes transposed convolutions to upsample the feature maps, reconstructing the segmented output. The dense blocks used in the skip connections play a crucial role in this process by merging high-resolution features from the VGG-16 encoder with the upsampled features in the decoder. This combination of features ensures that the segmentation output is both precise and detailed, effectively reconstructing the input image’s important attributes.

YSegNet 2: The YSegNet 2 model builds upon the architecture of YSegNet 1 by incorporating advanced features and modifications to enhance its performance. One of the key innovations in YSegNet 2 is its utilization of the output from YSegNet 1 as input to its encoder. This approach ensures that YSegNet 2 benefits from the refined and high-quality features extracted by YSegNet 1, leading to a more cohesive and continuous flow of information between the two models. By leveraging the features from YSegNet 1, YSegNet 2 can enhance its feature representation and achieve improved segmentation accuracy.

In addition to integrating features from YSegNet 1, YSegNet 2 incorporates attention gates within its skip connections. These attention gates are designed to focus on the most relevant spatial regions in the feature maps, filtering out less important information. The attention mechanism is applied to the skip connections between the encoder and decoder, allowing the model to selectively emphasize critical features while suppressing irrelevant background information. This targeted approach improves the precision of feature selection and enhances the overall quality of the segmentation output.

The integration of attention gates in YSegNet 2 works in conjunction with the features received from YSegNet 1, ensuring that the model effectively utilizes both the detailed information from the initial model and the refined features enhanced by the attention mechanism. This combination of inputs and advanced skip connections contributes to more accurate and reliable segmentation results, making YSegNet 2 a powerful tool for complex and detailed image analysis tasks.

The implementation of attention gates as skip connections in YSegNet 2 involves a systematic approach to refining feature processing and enhancing segmentation accuracy. Initially, the attention gate takes inputs from both the encoder and decoder. The decoder’s features serve as the gating signal, guiding the attention mechanism in determining which regions of the encoder’s features should be emphasized or suppressed. This is achieved by computing an attention map through convolutional layers followed by a sigmoid activation function, which produces values between 0 and 1. This map effectively highlights important features while attenuating irrelevant ones.

The refined encoder features are obtained by applying this attention map through element-wise multiplication, thereby weighting the features according to their relevance. These processed features are then passed to the decoder, where they are upsampled and combined with the decoder’s own features. This integration ensures that the decoder receives only the most pertinent information, enhancing its ability to reconstruct accurate and detailed segmentation outputs.

By focusing attention on relevant spatial regions and filtering out less important information, attention gates significantly improve the precision of the segmentation results. This mechanism allows YSegNet 2 to better utilize the features from both the encoder and decoder, resulting in a more refined and effective segmentation model.

Results and discussions

The dataset for experimentation is constructed by randomly selecting images from three datasets mentioned in “Introduction”. This approach is adopted to train the model with diverse types of ear images in various scenarios, thereby enhancing the model’s efficiency. The proposed dataset comprises a total of 3,140 images, with 440, 700, and 2,000 images sourced from the Human Face dataset, AMI dataset, and EarVN1.0 dataset, respectively. Due to variations in size among images from different datasets, all images are standardized by resizing them to a uniform size of 512 × 512 using the resize() function, an inbuilt tool in Python, with interpolation. The dataset is then randomly partitioned into training, validation, and testing sets in the ratio of 70:20:10, respectively. The training phase of the proposed experimentation involves approximately 50 epochs.

Performance metrics

Quantitative analysis is achieved using metrics derived from the confusion matrix (Shultz et al., 2011). The parameters considered are true positives (TP), true negatives (TN), false positives (FP) and false negatives (FN) as defined by Kumar et al. (2018). The evaluated metrics range from 0% to 1% or 0% to 100%, with the model’s efficacy increasing as the metric values rise.

(a) Accuracy: Segmentation accuracy refers to the alignment between the segmented maps of ground truth and the prediction (Csurka et al., 2013) and is expressed by Eq. (1).

(1) Accuracy=TP+TNTP+TN+FP+FN.

(b) Recall: The accurate detection of the foreground in relation to the overall ground truth foreground is termed recall (Sokolova, Japkowicz & Szpakowicz, 2006) or sensitivity, and its computation is defined by Eq. (2).

(2) Recall=TPTP+FN.

(c) Precision: Precision, alternatively recognized as positive predictive value, gauges the precision of accurately predicted foreground pixels in relation to the total number of segmented foreground pixels, and this is articulated through Eq. (3).

(3) Precision=TPTP+FP.

(d) Specificity: Specificity quantifies the accuracy of background prediction (Thanh, Prasath & Hien, 2019) and is determined using Eq. (4).

(4) Specificity=TNTN+FP.

(e) DICE score: DICE score also known as F1 score is calculated as the reciprocal of the average of the reciprocals of recall and precision (Kumar et al., 2018; Thanh et al., 2019b) and is given by Eq. (5).

(5) F1score=2×Recall×PrecisionRecall+Precision.

(f) IoU: The Intersection over Union (IoU) or Jaccard index metrics, as outlined in Thanh et al. (2019a), Taha & Hanbury (2015), represent the proportion of the intersection between elements in the true and predicted sets to the union of elements in these sets. This relationship is articulated in Eq. (6).

(6) IoU=DICE2−DICE.

Performance evaluation

Experimentation of the proposed framework is carried out on Google Colab, an open-source platform that offers a T4 GPU backend for Python programming. To enhance computational efficiency, the proposed ensemble model incorporates a pretrained VGG-16 as the initial encoder. The ensemble model is experimented with various competing encoder—decoder frameworks such as UNet, Double UNet (Jha et al., 2020), UNet++ (Zhou et al., 2018) and LS_YSegNet. Also, YSegNet with attention block as the skip connection represented as AG_YSegNet is considered for comparison. The proposed model is called as E_YSegNet (DANNET), where ‘E’ stands for ensemble. In the proposed work, YSegNet ensemble is used to enhance accuracy of segmentation in both constrained and unconstrained environment. This approach can help reduce errors and improve overall efficiency by considering multiple perspectives. The model undergoes a comprehensive evaluation encompassing both qualitative and quantitative analyses, with a qualitative comparison to existing methods presented in Fig. 3. The visual interpretation of segmented maps from the figure shows that in almost all cases the proposed model predicted the segmented maps effectively. All the edges of the segmented maps have a great similarity to the ground truth. Competing models were able to segment ear better in cases where the foreground ear occupied the major area of the image and performance of the models degraded in cases where the foreground was every small compared to the background. Due to the incorporation of dense bock, attention block and boundary extraction networks E_YSegNet outperformed other models in identifying the foreground and background correctly. The segmented maps of UNet and UNet++ was not effective even in cases where the foreground occupied the major area. This is because of the ambiguity or uncertainty in distinguishing between the foreground and background regions, especially if they share similar colours, textures, or structures. It is also noted that in the fifth case given in Fig. 3 the ear extension for the costume is also predicted by the competing models as ear. Whereas in the second case the boundary of the ear is not predicted desirably since some part of the ear is covered by hair. In such cases, E_YSegNet is able to predict efficiently as the model is trained based on the outputs from both the models. The validation accuracy and validation loss of LS_YSegNet, AG_YSegNet and E_YSegNet is plotted in Fig. 4 for 50 epochs. Validation accuracy and validation loss for E_YSegNet is highest and lowest, respectively, making it the most effective model for ear segmentation. Quantitative assessment of the proposed framework’s performance with other commonly used framewoks is tabulated in Table 1. The results indicate that the proposed ensemble model attains an overall accuracy of 98.93%. It can be seen that of the existing models, LS_YSegNet was only able to desirably predict foreground with a precision of 0.907 but AG_YSegNet outperformed with an increase of 2.4% in precision. The proposed ensemble performed remarkably with an increase of 5.4% in precision. Further, the proposed E_YSegNet improves the recall rate by 6.4%, while the DICE score and IoU parameters are increased by 5.9% and 10.8%, respectively. Finally, the execution time for each experimented model is analysed. The training and testing durations are outlined in Table 2. When training and testing models in Python using TensorFlow, execution time can be measured to assess performance. The suggested model exhibits the longest training duration due to its depth and complexity, while UNet demands the least time owing to its relative simplicity compared to other models. Additionally, it is noted that the testing time for the proposed model is significant. Consequently, the performance analysis indicates that the ensemble model incorporating a boundary extraction network outperforms others in obtaining a segmented map. This positions the proposed model as a valuable tool for ear detection across diverse applications.

Figure 3 Qualitative performance comparison of output with ground truth.

Figure 4 Performance evaluation based on validation accuracy and loss.

Table 1 Comparisons of quantitative metrics assessing the performance of the ear segmentation.

This table represents the qualitative and quantitative performance metrics of the proposed method.

Model							
Metrics	U-Net	Double U-Net	UNet++	LS_YSegNet	AG_YSegNet	Proposed E_YSegNet	
Accuracy	84.37%	89.76%	90.74%	91.53%	93.43%	98.93%	
Recall	0.837	0.852	0.868	0.893	0.917	0.981	
Precision	0.803	0.837	0.852	0.907	0.931	0.985	
Specificity	0.816	0.841	0.883	0.913	0.903	0.974	
DICE score	0.82	0.844	0.872	0.90	0.924	0.983	
IoU	0.695	0.73	0.773	0.81	0.859	0.967	

Table 2 Execution time of experimented frameworks comparison.

This table represents the calculation of performance metric such as execution time.

Framework	Learning duration (s)	Evaluation duration (s)	
UNet	386	3.8	
Double UNet	523	5.2	
UNet++	752	6.8	
LS_YSegNet	543	5.4	
AG_YSegNet	552	6.1	
E_YSegNet	1273	6.5	

Conclusion

The article presented a deep ensemble consisting of two YSegNet models, one with a dense block and the other with an attention block as their skip connection. This model is proposed to segment ears from facial images, which can be useful for ear biometrics and other relevant applications. The work aims to ensure efficient performance regardless of the type of input facial images by building a new dataset for experimentation, combining images from three different datasets: AMI, EarVN1.0, and Human Face. Additionally, the proposed approach’s performance is evaluated by assessing the accuracy of predictions against the corresponding ground truth. The proposed model was compared with other competing encoder-decoder models such as UNet, Double UNet, and UNet++, as well as LS_YSegNet and AG_YSegNet, using the developed dataset. Although all the models performed reasonably well overall, only the proposed model was able to accurately detect and segment ears from images with numerous objects. The experimental findings demonstrate that the ensemble YSegNet with dense blocks and attention blocks effectively and efficiently segments ears and their edges, even in complex images. LS_YSegNet is able to identify the boundary of the ears, while AG_YSegNet develops effective segmented maps based on the backscattering coefficient. Consequently, the ensemble model achieved an accuracy of 98.93%, with significant improvements in evaluation metrics such as precision, recall, DICE score, and IoU.

The implications of this research extend to various domains where accurate ear segmentation is crucial, such as biometric identification and security. By improving the robustness and accuracy of ear detection, the proposed model enhances the reliability of biometric systems, particularly in scenarios where facial features are obscured by masks or other coverings. However, the proposed ensemble model has certain limitations, such as increased computational complexity and a potential risk of overfitting with limited datasets, which could impact its scalability and real-time applicability. Future work could address these limitations and further improve the model’s efficiency by experimenting in real-time and on a larger variety of datasets. Additionally, the proposed model could be explored in specific domains where accurate foreground segmentation is crucial, such as medical imaging, autonomous driving, satellite imagery analysis, and surveillance. These advancements could lead to broader applications and more refined biometric identification systems.

Supplemental Information

Supplemental Information 1 Code.

Supplemental Information 2 Eardataset.

The EarVN1.0 dataset is a comprehensive collection of over 28,412 ear images from 164 individuals. It encompasses a wide range of variations in pose, scale, illumination, occlusion, resolution, and lighting conditions. This dataset is suitable for various applications such as person authentication and classification.

This file contains a sample subset of EarVN1.0 that consists of selected ear images from male participants (sampled from Person IDs 1–98). The images capture a variety of angles, lighting conditions, and backgrounds to ensure diversity and support robust model training for male ear recognition tasks.

Supplemental Information 3 Eardataset.

This file contains a sample subset of EarVN1.0 that includes selected ear images from female participants (sampled from Person IDs 99–164). The images are chosen to represent variations in pose, illumination, and environmental settings, offering a comprehensive foundation for female ear recognition studies.

Supplemental Information 4 Evaluation of the Execution Time used in the Proposed Method for implementation.

Additional Information and Declarations

Competing Interests

Author Contributions

Data Availability

Natalia Kryvinska is an Academic Editor for PeerJ.

Deepthy Mary Alex conceived and designed the experiments, performed the experiments, analyzed the data, performed the computation work, prepared figures and/or tables, authored or reviewed drafts of the article, and approved the final draft.

Kalpana Chowdary M. conceived and designed the experiments, performed the experiments, analyzed the data, performed the computation work, prepared figures and/or tables, authored or reviewed drafts of the article, and approved the final draft.

Hanan Abdullah Mengash conceived and designed the experiments, performed the experiments, prepared figures and/or tables, and approved the final draft.

Venkata Dasu M. conceived and designed the experiments, performed the experiments, analyzed the data, performed the computation work, prepared figures and/or tables, authored or reviewed drafts of the article, and approved the final draft.

Natalia Kryvinska conceived and designed the experiments, analyzed the data, prepared figures and/or tables, authored or reviewed drafts of the article, and approved the final draft.

Chinna Babu J. conceived and designed the experiments, prepared figures and/or tables, authored or reviewed drafts of the article, and approved the final draft.

Ajmeera Kiran conceived and designed the experiments, performed the experiments, performed the computation work, authored or reviewed drafts of the article, and approved the final draft.

The following information was supplied regarding data availability:

The AMI Ear Database is available at https://webctim.ulpgc.es/research_works/ami_ear_database/#whole.

The EarVN1.0 dataset is available at Mendeley: Truong Hoang, Vinh (2020), “EarVN1.0”, Mendeley Data, V4, doi: 10.17632/yws3v3mwx3.4.

The Human Face (Ear Detection with Annotation) dataset is available at Kaggle: https://www.kaggle.com/datasets/harshghadiya/human-face-ear-detection-with-annotation.

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
