# Peer review of "DANNET: deep attention neural network for efficient ear identification in biometrics"

_PeerJ Computer Science, doi:10.7717/peerj-cs.2603_

## Round 0.1 · original submission · Major Revisions

After carefully considering the reviews and assessing your manuscript, I am pleased to inform you that we would like to invite you to revise and resubmit your manuscript for further consideration. The reviewers have provided constructive comments that will help strengthen your work. Although, one of the reviewers is not in favor, however, others find some merits. Please address each of these points thoroughly in your revised manuscript. Additionally, ensure that you provide a detailed response letter outlining how you have addressed each comment raised by the reviewers. This will help the reviewers and myself to evaluate the changes made to the manuscript. It is PeerJ's policy that additional references suggested during the peer-review process should only be included if the authors agree that they are relevant and useful. Good luck.

·

Basic reporting

1- The writing in this manuscript is hard to understand because of mistakes with grammar and phrasing. To make it clearer, it would be helpful to have someone who speaks English perfectly or a professional editor review it.

2- The introduction doesn't explain well enough why this research is important. It mentions that ear biometrics might be useful because people wear masks now due to COVID-19, but it doesn't explain why this method would be better than other ways we already have to identify people.

3- This research mentions a lot of other studies, but it doesn't explain how those studies connect to this new research. The writing would be stronger if it explained what past research has already found and how this new work builds on that or does something different and better.

4- it's hard to follow because the different sections don't connect smoothly. The transitions between ideas are jumpy, and it's like reading bits and pieces instead of a whole story. To make it easier to read and understand, the order of the information would need to be reorganized. It's like reading separate paragraphs instead of one connected story. To make it easier to understand, the information needs to be rearranged in a more logical order.

5- The pictures (figures) are clear, but they don't always help explain the text. Some pictures seem unnecessary, and a few don't have clear labels or captions that explain what they're showing.
Examples:
5.1- Figure 3: The structure of LS_YSegNet is shown, but there is no detailed explanation in the text linking the figure to the discussion on LS_YSegNet, making its inclusion seem redundant.
5.2- Figure 6: The schematic representation of the proposed model is complex and lacks clear labels and captions, making it difficult for readers to understand its relevance without additional context.
5.3- Figure 8: Sample images from the proposed dataset are presented, but their inclusion does not add significant value to the manuscript. The text does not adequately explain why these specific images are important or how they were selected.

Experimental design

1- The research presents itself as original but does not convincingly establish its novelty within the existing body of work. The manuscript lacks a strong narrative on how it significantly differs from or improves upon previous studies.

2- The writing doesn't explain what problem it's trying to solve or what new information it's trying to find. Because of this, it's hard to see why this research is important.

3- the methods section isn't detailed enough. The writing doesn't explain exactly how they trained their model or the specific settings they used for the computer program (neural network).

Validity of the findings

1- The writing doesn't discuss how important or new the findings might be. It also doesn't explain how this research fits in with other scientific work in this field.
2- Even though the raw data is provided, it would be hard for other researchers to repeat this study because the methods section is not detailed enough and the reasons for doing things a certain way are not clear.
3- It's hard to know if the results are reliable because the writing doesn't explain the data analysis well enough. There aren't enough details about how they checked the data or made sure it was accurate.
4- The conclusions don't seem to be based strongly on the research questions or the results. They seem too positive and don't consider the weaknesses of the study or how the results might be biased.

Additional comments

Weaknesses:
1- The manuscript is riddled with grammatical errors and awkward phrasing.
2- The research question is poorly defined, and the rationale for the study is weak.
3- The methods section lacks sufficient detail for replication.
4- The discussion and conclusions are superficial and do not critically engage with the findings.

Specific Suggestions for Improvement
1-The introduction needs a stronger justification for the study. Expand on the knowledge gap being filled and provide a more compelling rationale for why this research is necessary.
2- The manuscript should undergo a thorough review for grammatical errors and awkward phrasing. Examples include lines 23, 77, 121, 128.
3- Provide detailed descriptions of the experimental setup, including the implementation of attention gates and their specific role in the proposed model.
4- Expand the discussion on the limitations of the proposed model and potential areas for future research. Address any biases or challenges encountered during the study.

Reviewer 2 ·

Basic reporting

The paper submitted here reflects Ear Identification in Biometrics using Neural Networks. The paper is poorly written in terms of English and Vocabulary. Several Grammer issues are observed that the authors did not take care of.
The abstract is poorly written without any specification of the concept and its applicability. The authors failed to correlate with facts about the need for Ear Biometrics. There is no scientific correlation in selecting the topic for biometrics identification. They fail to satisfy the urge to use ear biometrics with face or fingerprint identification. Also, the era of COVID-19 is now over, so there is no requirement to find additional biometrics identifications for the human race. Accurate detection or identification of the ear is indeed the biggest challenge that the study's authors do not manage well. The introduction is poorly written with the latest references and citations. In the introduction, there are missing references for terms such as Canny Detection. There is no State-of-the-art comparison for the similar models proposed by other researchers.
Section 2 is not required in the article. The authors' ordering of the sections is not correct, which makes it understandable.
The authors fail to understand a proper reason for selecting Related works in the proposed model. They have not provided any justification for "Why" these models were chosen. This weakens the choice of the inspirational works that helped the authors to move ahead for the proposed model.

Experimental design

"A novel deep YSegNet with a combination of dense block and attention gates as skip connection is proposed to obtain segmented output maps of ears", as proposed by the authors. However, they are still unable to justify why these Attention Gates are used for segmentation with YSegNet.
The bias vector and filter size are not chosen with a scientific explanation of the choice for selection. Equation 2 terms are not explained in detail so that readers can understand eligibility for using the equation.
Equations 3,4,5,6,7,8 do not need to be explained and placed in section 5.1 since they are the basics of any ML model. The authors should not explain general terms to enhance the length of the paper.
Three datasets are trained for the experiment; however, only two data sets are provided in supplementary files. This fails to understand their results.
"The training phase of the proposed experimentation involves approximately 50 epochs", as proposed by the authors. However, a selection of 50 epoch size is not mentioned for the training.

Validity of the findings

"The proposed ensemble model incorporates a pre-trained VGG-16 as the initial encoder"; however, the mechanism for the Pre-Training of the VGG-16 encoder is not explained. Also, the manuscript's integration of Google Colab with the encoder input is not explicit.
The authors do not adequately mention the selection of Competing Encoders to ensure the proper comparison. They also fail to compare the latest encoders with the present model VGG-16 encoder.
There is no explanation relative to the dataset's training and testing in the model proposed in the study, and the authors are also unable to justify the use of datasets in the Python Code.
The code in the Python file uses DenseNet, which is not mentioned anywhere in the paper. There is still confusion about using the libraries in the Python Program.
Even the Google Colab File:
https://colab.research.google.com/drive/1oJuJNSsjWBkLGYHabpmkRXWTxW2jECJr
is not under a free access / open source license for public use. This makes it difficult for readers to navigate the code provided in the section.

Additional comments

Overall, the model presented in this study fails to justify the applicability adequately and is not recommended for publication. There can be better options to explore and work with to strengthen the paper. The authors are required to enhance the quality of the article and manage the complete manuscript in a new way. The acceptance in the present form is highly not recommended. The above findings should enhance the manuscript to make it more solid and relevant for publication. In the present format, the manuscript lacks novelty in research, experimenting errors, invalidated outcomes and probably incorrect results. They authors must modify the article to make it more strong.

Reviewer 3 ·

Basic reporting

The abstract should be revised. The abstract is insufficient in terms of containing the results.
Information about the general structure of the study is complex and insufficient.
The conclusion section should be revised and supported with the implications and future work.
The English should be checked and also checked for grammar and typos.
The motivation, contribution, and benefits should be the part of introduction section.
how your research is different from others?
the explanation of each equation should be mentioned.

Experimental design

How are the parameters used for algorithms determined? Has a Preliminary Study been conducted?
The effects of the results of the study need to be analyzed in detail.
On what basis was the performance evaluation made? What metrics were used? Why is only the accuracy rate shown? There is no detailed information about these.
The conclusion section is very inadequate. The results of such a study need to be better conveyed to the reader.

On what basis was the performance evaluation made? What metrics were used? Why is only the accuracy rate shown? There is no detailed information about these.

Validity of the findings

The conclusion section is very inadequate. The results of such a study need to be better conveyed to the reader.
How are the parameters used for algorithms determined? Has a Preliminary Study been conducted? Or do they take from another paper? They need to be more clear about that.
The effects of the results of the study need to be analyzed in detail. They should add a discussion section.
please add the discussion part before the conclusion section. please mention the key findings of this study.

Reviewer 4 ·

Basic reporting

In this paper, a sophisticated method was proposed, named as an encoder-decoder deep learning ensemble technique incorporating attention blocks. Experimental validation utilizes a combination of data from EarVN1.0, AMI, and Human Face datasets, achieving a segmentation framework accuracy of 98.93%. The proposed innovative method demonstrates potential for individual recognition, particularly in large gatherings, when complemented by an effective surveillance system.

Experimental design

The abstract is too short, more detailed information needs to be given.
They should put a flow chart about the proposed model, it is insufficient as it is.
Experimental studies are not understandable and sufficient for the reader. What is its difference from studies in the literature? Why should we choose this model? These are not understandable to the reader.

The paper is generally well thought out, but the structure of the paper is quite poor. It needs to be rewritten from scratch. It is not understandable in this situation.

Validity of the findings

The abstract is too short, more detailed information needs to be given.
They should put a flow diagram about the proposed model, it is insufficient as it is.
Experimental studies are not understandable and sufficient for the reader. What is its difference from studies in the literature? Why should we choose this model? These are not understandable to the reader.

The paper is generally well thought out, but the structure of the paper is quite poor. It needs to be rewritten from scratch. It is not understandable in this situation.

Additional comments

The abstract is too short, more detailed information needs to be given.
They should put a flow diagram about the proposed model, it is insufficient as it is.
Experimental studies are not understandable and sufficient for the reader. What is its difference from studies in the literature? Why should we choose this model? These are not understandable to the reader.

The paper is generally well thought out, but the structure of the paper is quite poor. It needs to be rewritten from scratch. It is not understandable in this situation.

---

## Round 0.2 · Major Revisions

Both are reviewers are not satisfied with the revised manuscript. Many of their concerns were not properly addressed. They have provided very constructive suggestions and recommendation a major revision. Authors are required to go through each and every comments thoroughly, revised manuscript inline with the suggestions, and prepare a point-by-point rebuttal. Good luck

Reviewer 2 ·

Basic reporting

The paper still requires significant improvement in English grammar and comprehension.
Professional proofreading services should be engaged to ensure the paper's language is of high quality. Collaborating with native English speakers could also be beneficial.

The figures submitted are unclear and do not follow the journal guidelines. (Specially figure 4,5,7,8)

The paper's structure could be improved by merging the introductory section with the literature survey or enhancing it with more relevant information. This would help ensure a smoother flow of information from the first to the last section. The literature survey section is also poorly designed and managed. Four case studies are mentioned; however, the section needs spelling correction. The authors have not adequately revised the language.

The existing open source year data set description is optional as a single section. It should be merged with the methodology or disclosed as a subsection. Although the authors were notified in the previous edition to work ahead with it, they still need to section the article appropriately.

There is also a subsection on related works, which should also be presented in the literature review. The previous revision also noted this, but the sub-sectioning still needs to be thoroughly done. The authors are advised to reconsider the proper positioning of their sections, and it would be much better if they could represent the sections with a flowchart or a diagram.

Experimental design

In section three, the authors discuss the three types of related frameworks. However, they do not mention the relevance of the presence. There is no need for explanations, and significance is not used in the article for the research.

While equation one addresses the convolution parameters,  the need for integration in the present study is a pressing issue that remains fully understood.

Attention gates are mentioned to remove irrelevant information from the encoder and decoder layers. However, the proper mechanism for doing so is not reflected, which is one of the most important parts of this study.

This study considers linear transformation for simplicity. However, the results generated with linear transformation are not accurate. The authors are advised to update the sampling method for the transformation to provide proper results. The transformation required should be done with proper mathematical equations and a new structure of the experimental analysis.

Only new citations and references are introduced in the study without updating the revision suggested in the previous round.

The authors claim that 3140 images were distributed in a 70:20:10 ratio. The validation training and testing are done on the uniform image size of 512X 512. However, the data sets comprise images of varying lengths. The need to be used to correct the images from these data sets should be mentioned. They mentioned that data sets in section 3 contain images that are very small or very large in resolution. The pixel resolution is unified, with some processes that still remain hidden. The authors must incorporate the proper reasoning of image correction. Since images are one of the most crucial parts of testing and training the model proposed, it is mandatory to comply with standardized image formats and resolutions.

Validity of the findings

Table 1 and Table 2 are both named TABLE 1 in the figure caption. This is not acceptable.
In the previous revision, the figure legend and captions were mentioned as needing to be checked along with their proper placement in the text. However, it appears that the authors have yet to complete that.

The manuscript does not explain experimentally the comparisons of quantitative metrics assessing the performance of ear segmentation. It does not mention their execution and arrangement in the experimental analysis. The manuscript should explain the findings of the State-of-the-art models, and then a comparison is needed. Authors are advised to update the comparison and mention the complete execution procedure in this comparison to make their manuscript more acceptable.

The execution time of the experimented frameworks is mentioned in a table, but there is no information related to the time calculation. The code supplied in the supplementary file is insufficient to support the calculation of the time parameter. The authors must supply proper code to ensure the time calculation in this case.

Additional comments

The authors use obsolete references in the study. Some more recent references are expected instead. The authors are advised to modify their references and include more relevant ones.

Traces of the Code submitted by the authors are found in:
https://github.com/ahukui/BOWDANet/blob/master/Layers.py
Either the authors must mention this in the manuscript, or they should develop their code to ensure the novelty of the experiment.

The article must be rechecked and updated to ensure the novelty of the research and proper experimental association.

Reviewer 3 ·

Basic reporting

I did not see a clear big change in the revised version. The authors suggested providing a point-point reply to each reviewer's comments instead of writing;

"The abstract highlights accuracy as it is a key performance metric that directly reflects the model's effectiveness in correctly identifying and segmenting ears, demonstrating the superior performance of the ensemble YSegNet model over other methods. Including a single, impressive metric like accuracy makes the abstract concise and impactful, quickly conveying the model's success to the reader. "

I suggest you please reply to each comment and also mention specific paragraph line number and page number. The revised paper should be highlighted in yellow color.

* In the revised version the introduction is very short not acceptable. Please modify as per the previous comments.
* The paper organization is missing. Please add it at the end of the introduction section.

Experimental design

* why two related work sections? I don't understand.
* Please update the manuscript based on previous comments.

Validity of the findings

* Please update the manuscript based on previous comments.

---

## Round 0.3 · Major Revisions

The reviewers did not find any significant improvement in the revised manuscript. You are asked to revise your manuscript another time.

Reviewer 2 ·

Basic reporting

I did not see an apparent significant change in the revised version.
The authors suggested providing a point-point reply to each reviewer's comments instead of writing them.

The English language upgrade is still not achieved.
Proper proofreading is not done, and the language still lacks the required standard.

The authors have not adequately justified the prescribed changes needed in the manuscript.
They should highlight the line numbers in the text about the update and their relevance.
A proper comment per suggestion should be produced.
The updated revision seems to be more amalgamated due to the complex structuring.
The authors should recheck the modifications and restructure the manuscript to reflect the proper change.

The paper seems highly unorganized, and even the previous suggestions seem undiscovered. The authors must take it seriously and provide some novel research-based comprehensive inputs to ensure that the manuscript contributes to the required outcomes.

Experimental design

The experimental design is still not available.
The collab file for the code is not available for public access.
The authors submitted a demo code for the time calculation per the review. However, the code is not intended and submerged in the experimental setup.
The authors are still expected to update the entire code and fork it to a public repository like GitHub.
The experimental analysis is impossible unless the complete Collab Files and integration of the Time calculation are done. Even the results for the same are required to be integrated.

There is still a significant gap in the revision expected in the experiment.
The authors have not revised the needful.
The datasets are obsolete, and no new additions are made from the experimental inputs.

Validity of the findings

The experimental setup should be reconsidered to obtain proper results.
Using an actual dataset instead would be a great approach to ensure better results.

Still, there is an enormous scope to resolve the previously stated comments.
The outcomes in the time calculations are not integrated into the actual code.
The authors are advised to incorporate the changes in the code instead of only executing the pre-existing code.

Additional comments

Significant revisions are still pending from the authors' side.
The revision of the English Language, proper datasets, code revision, code uploading, results calculation, time integration, etc., must be completed.
The authors have not yet accomplished the previous comments.
The manuscript still lacks novelty and profound results that would enable it to be published in a reputed journal.
The authors are advised to pay special attention to the revision in this round to ensure the required updates are made.

Reviewer 3 ·

Basic reporting

The reviewrs modified paper as per suggestions. However, still it has english and grammer mistakes. Plesae modify in the final version.

Experimental design

The reviewrs modified paper as per suggestions.

Validity of the findings

The reviewrs modified paper as per suggestions.

---

## Round 0.4 · Minor Revisions

(1) There are some minor typos that need to be corrected.
(i) For instance, Line No. 44, "COVID" should be "COVID-19".
(ii) Line No. 75, CNN should be expanded on its first occurence.
(iii) Line No. 232, Is "resize ()" a tool? It seems to be a function. Pl verify and correct.
(2) The contribution statement at the end of the Introduction section is not clear. Authors need to mention the contribution made in this paper.
(3) Fig. 1 is irrelevant and must be dropped.
(4) In the title, the proposed method has been named as "DANNET". However, in manuscript, it is named as "E_YSegNet". Authors need to write it uniformly throughout the manuscript.
(5) The experimental design section seems irrelevant. However, "3.2 experimental setup" section is important. I would suggest to merge experimental setup section with "Results & Discussion", where author may first describe the experimental setup.
(6) Limitation and future direction must be mentioned in the conclusion section.

Reviewer 2 ·

Basic reporting

The authors have addressed the concerned issues in the revision.
The article is acceptable in its present form.

Experimental design

The authors have addressed the concerned issues in the revision.
The article is acceptable in its present form.

Validity of the findings

The authors have addressed the concerned issues in the revision.
The article is acceptable in its present form.

Additional comments

The authors have addressed the concerned issues in the revision.
The article is acceptable in its present form.

---

## Round 0.5 · accepted · Accept

I am pleased to inform you that your paper has been accepted for publication in PeerJ Computer Science. Your manuscript has undergone rigorous peer review, and I am delighted to say that it has been met with praise from our reviewers and editorial team. Your research makes a significant contribution to the field, and we believe it will be of great interest to our readership. On behalf of the editorial board, I extend our warmest congratulations to you.